



# Summer fluxes of methane and carbon dioxide from a pond and floating mat in a continental Canadian peatland

M. Burger[1,2], S. Berger[1,2], I. Spangenberg[1,2] and C. Blodau[1,2]

[1] Ecohydrology and Biogeochemistry Group, Institute of Landscape Ecology, University of Münster, Germany

[2] School of Environmental Sciences, University of Guelph, Canada

*Correspondence to*: C. Blodau (christian.blodau@uni-muenster.de)

**Abstract**

Ponds smaller than 10000 $m^2$ likely account for about one third of the global lake perimeter. The release of methane ($CH_4$) and carbon dioxide ($CO_2$) from these ponds is often high and significant on the landscape scale. We measured $CO_2$ and $CH_4$ fluxes in a temperate peatland in southern Ontario, Canada, in summer 2014 along a transect from the open water of a small pond (847 $m^2$) towards the surrounding floating mat (5993 $m^2$) and in a peatland reference area. We used a high-frequency closed chamber technique and distinguished between diffusive and ebullitive $CH_4$ fluxes. $CH_4$ fluxes and $CH_4$ bubble frequency increased from a median of 0.14 (0.00 to 0.43) mmol $m^{-2}$ $h^{-1}$ and 4 events $m^{-2}$ $h^{-1}$ on the open water to a median of 0.80 (0.20 to 14.97) mmol $m^{-2}$ $h^{-1}$ and 168 events $m^{-2}$ $h^{-1}$ on the floating mat. The mat was a summer hot spot of $CH_4$ emissions. Fluxes were one order of magnitude higher than at an adjacent peatland site. During daytime the pond was a net source of $CO_2$ equivalents to the atmosphere amounting to 0.13 (−0.02 to 1.06) g $CO_2$ equivalents $m^{-2}$ $h^{-1}$, whereas the adjacent peatland site acted as a sink of −0.78 (−1.54 to 0.29) g $CO_2$ equivalents $m^{-2}$ $h^{-1}$. The photosynthetic $CO_2$ uptake on the floating mat did not counterbalance the high $CH_4$ emissions, which turned the floating mat into a strong net source of 0.21 (−0.11 to 2.12) g $CO_2$ equivalents $m^{-2}$ $h^{-1}$. This study highlights the large small-scale variability of $CH_4$ fluxes and $CH_4$ bubble frequency at the peatland-pond interface and the importance of the often large ecotone areas surrounding small ponds as a source of greenhouse gases to the atmosphere.



## 1. Introduction

Inland waters play a significant role in the global carbon cycle although covering only 3.7 % of the Earth's land surface (Bastviken et al., 2011; Raymond et al., 2013; Tranvik et al., 2009). They transport and sequester autochthonous and terrestrially derived carbon and are also sources of carbon dioxide ($CO_2$) and methane ($CH_4$) to the atmosphere (Cole et al., 2007; Tranvik et al., 2009). Global estimates of $CO_2$ and $CH_4$ emissions from inland waters have recently been corrected upward to 2.1 Pg C $yr^{-1}$ as $CO_2$ (Raymond et al., 2013) and 0.65 Pg C $yr^{-1}$ as $CH_4$ (Bastviken et al., 2011). Together they are similar to the net carbon uptake by terrestrial ecosystems of $-2.5 \pm 1.3$ Pg C $yr^{-1}$ and to approximately one third of the anthropogenic $CO_2$ emissions (Ciais et al., 2013).

Small aquatic systems may be particularly important in this respect (Downing, 2010). According to high-resolution satellite imagery analyzed by Verpoorter et al. (2014), 77 % of the total 117 million lakes belong to the smallest detectable size category of 2000 to 10000 $m^2$ lake area. These waters only contribute 7 % to the area but 32 % to the total lake perimeter (Verpoorter et al., 2014). Numerous processes were found to proceed faster in small aquatic systems than in larger ones. Sequestration rates of organic carbon (Downing, 2010; Downing et al., 2008), the concentrations of $CH_4$, $CO_2$, and dissolved organic carbon (DOC) in the water column (Bastviken et al., 2004; Juutinen et al., 2009; Kelly et al., 2001; Kortelainen et al., 2006; Xenopoulos et al., 2003), and $CH_4$ and $CO_2$ emissions from the water to the atmosphere increase with decreasing lake size (Juutinen et al., 2009; Kortelainen et al., 2006; Michmerhuizen et al., 1996; Repo et al., 2007).

Small and shallow lakes and ponds are common in flat northern glacial landscapes and abundant in peatland areas, where 20 to 30 % of the world's soil organic carbon is stored (Turunen et al., 2002). $CO_2$ emissions from peatland ponds were reported to be in the same order of magnitude than net uptake of $CO_2$ by the peatland vegetation (Dinsmore et al., 2009; Hamilton et al., 1994). $CH_4$ emissions from open waters generally exceed $CH_4$ fluxes from vegetated areas by a factor 3 to 25 (Hamilton et al., 1994; McLaughlin and Webster, 2014; Trudeau et al., 2013). Moreover, $CH_4$ and $CO_2$ emissions from open waters can be significant on the landscape scale despite their often small area (Dinsmore et al., 2010; Juutinen et al., 2013). Pelletier et al. (2014) estimated that a pond cover of $> 37$ % could convert a northern peatland from a carbon sink into a carbon source. Such findings are relevant as Hamilton et al. (1994) and Trudeau et al. (2013) reported a pond cover of 8 to 12 % and 42 % in fens and bogs in northern Canada. The authors suspected a contribution of aquatic $CH_4$ fluxes to landscape $CH_4$ fluxes of 30 % and 79 %, respectively. Very high $CH_4$ emissions have also been reported from a floating mat on a thermokarst pond and a floating mat within a bog pond (Flessa et al., 2008; Sugimoto and Fujita, 1997). Juutinen





et al. (2013) documented highest CH$_4$ fluxes from a wet lawn adjacent to a small fen lake
compared to the lake itself and fen lawns farther away from the small lake.
Fluxes of CH$_4$ and CO$_2$ from ponds are controlled by environmental and biotic factors.
Atmospheric CH$_4$ fluxes are controlled by microbial production and oxidation of CH$_4$ within
peat, sediment and surface water and the diffusive, ebullitive, and plant-mediated transport to
the atmosphere (Bastviken et al., 2004; Bridgham et al., 2013; Carmichael et al., 2014). CO$_2$
exchange is driven by the interplay of heterotrophic and autotrophic respiration and by
photosynthesis of aquatic macrophytes and algae. Both gas fluxes are linked to the quantity and
quality of organic and inorganic carbon supplied from the surrounding catchment (Huttunen et
al., 2002; Macrae et al., 2004; Tranvik et al., 2009). They are also related to temperature, wind
speed and air pressure (e. g. Trudeau et al., 2013; Varadharajan and Hemond, 2012; Wik et al.,
2013). Ebullition appears to be of particular importance for CH$_4$ release to the atmosphere
(Walter et al., 2006; Wik et al., 2013) and varies on scales of several tens to hundreds of meters
(Bastviken et al., 2004; Wik et al., 2013). Emissions of CH$_4$ emissions are generally lower in
the pelagic than in the littoral zone, where plant habitats further influence fluxes (Juutinen et
al., 2001; Larmola et al., 2004). On the other hand, Trudeau et al. (2013) found 2.5 to 5 times
lower CH$_4$ fluxes at the border of fen pools than in the center of the pools with areas of 60 and
200 m$^2$.
Despite this progress, knowledge on the temporal and spatial variability of CH$_4$ and CO$_2$ fluxes
within small pond systems is limited. We know, for example, little about the CH$_4$ and CO$_2$
exchange of transition zones between ponds and surrounding peatlands, which can be especially
important due to the high perimeter to area ratio of small ponds (Verpoorter et al., 2014). It is
important to consider the net effect of different microforms of peatlands by taking into account
the global warming potentials, as CH$_4$ emissions may easily offset carbon sinks in ponds. To
gain more insight into these issues we investigated the summer atmospheric CO$_2$ and CH$_4$
exchange of open water, a floating mat and an adjacent peatland area in a temperate peatland
in southern Ontario, Canada. In particular we tested the hypothesis that (I) ebullitive and
diffusive CH$_4$ fluxes increase from the open water towards a floating mat surrounding the pond.
We examined (II) if CH$_4$ and CO$_2$ effluxes from the system increases with temperature and
wind speed and investigated if falling air pressure raises CH$_4$ fluxes. To assess the importance
of the pond system for the greenhouse gas balance we calculated the net radiative forcing of the
investigated peatland microforms.



**2 Materials and methods**
**2.1 Study site**
Wylde Lake Bog is located in the southeastern part of the Luther Marsh Wildlife Management
Area (43°54.667' N, 80°24.022' W) (Fig. 1) at about 490 m above sea level and has an area of
approximately 7.8 km$^2$. A 600 cm deep profile analyzed by Givelet et al. (2003) documented
clay-rich sediments up to 560 cm depth, gyttja from 560 to 490 cm, fen peat from 490 to
approximately 300 cm and bog peat above 300 cm depth. The peatland is dominated by mosses,
graminoids, dwarf shrubs and sporadic trees, and a pronounced hummock-hollow-
microtopography. Common in the peatland are *Sphagnum magellanicum*, *S. capillifolium*,
*Carex disperma* and *Chamaedaphne calyculata* and on the floating mat *S. angustifolium, S.*
*magellanicum* and *Rhynchospora alba*. The plant species composition of the study site is given
in the Supplementary Information (Table S1). The vicinity of the pond is characterized by
small open and larger treed areas dominated by *Larix laricina* and *Picea mariana*. The pond
(Fig. 1) has an area of 847 m$^2$ and a depth of 0.3 to 0.8 m. The interface between the water
column and the organic deposits is not clearly delimited but consists of a transition zone with
suspended organic material. It likely has changed in size, depth, and shape throughout the last
decades. Sandilands (1984) reported that larger, adjacent Wylde Lake shrunk from 0.4 km$^2$ in
1928 to 0.05 km$^2$ in 1984. The floating mat (Fig. 1) surrounding the pond has an area of approx.
5993 m$^2$. Climate is temperate continental with a mean annual air temperature of about 6.7 °C,
annual precipitation of 946 mm including 148 mm of snowfall, and an average frost-free period
from May 7$^{th}$ to October 6$^{th}$ (1981 to 2010, Fergus Shand Dam, National Climate Data and
Information Archive, 2014).
**2.2 Environmental variables**
Air temperature, relative humidity, wind speed, wind direction, photosynthetically active
radiation (PAR) and precipitation were recorded at the study site by a HOBO U30 weather
station (U30-NRC-SYS-B, Onset) (Supplementary Information, Table S2). Water temperature
of the pond and the temperature of the floating mat were also continuously measured. Air
pressure was recorded at a distance of 1.1 km from the study site (Supplementary Information,
Table S2).
**2.3 CH$_4$ and CO$_2$ flux measurements with closed chambers**
CH$_4$ and CO$_2$ fluxes of the pond and the floating mat were measured once a week from July
10$^{th}$ to September 29$^{th}$, 2014 between 1 pm ± 1.5 hours and 5 pm ± 1.5 hours using closed
chambers designed according to Drösler (2005). The cylindrical, transparent Plexiglas



chambers had a basal area of 0.12 $m^2$ and a height of 0.40 m. They were equipped with 2 or 3
fans (Micronel Ventilator D341T012GK-2, BEDEK GmbH) to circulate the air, a
photosynthetically active radiation and an air temperature sensor (Supplementary Information,
Table S2). To compensate for air pressure differences, we attached a vent tube, 12 cm long and
7 mm inner diameter, to the chamber (Davidson et al., 2002). Transparent chambers were used
to measure net ecosystem exchange (NEE) and cooled with up to 6 ice packs depending on
ambient temperature to ensure a temperature change of less than 1°°C during the chamber
closure. Ecosystem respiration (ER) was measured with chambers covered with reflective
insolation foil. On the water, chambers were operated with a Styrofoam float (0.80 m × 0.61 m
× 0.08 m). The chamber walls extended 10 cm below the water surface as recommended by
Soumis et al. (2008). $CH_4$ and $CO_2$ concentrations were quantified with an Ultraportable
Greenhouse Gas Analyzer (915-001, Los Gatos Research) at a temporal resolution of 1 s.
According to the manufacturer, a single data point has a precision of < 2 ppb for $CH_4$ and
< 300 ppb for $CO_2$. Stability of the calibration was checked in March and August 2014. The air
was circulated between the chamber and the analyzer through low-density polyethylene tubes
of 5 m length with an inner diameter of 2 mm and a water vapor trap. Using this setup it took
36 s until the sampling cell of the analyzer was fully flushed and the concentration had
stabilized.
Flux measurements on the open water were carried out in 6 locations with increasing distance
of 0.7 m to 4.6 m to the floating mat (Supplementary Information, Table S3). On the floating
mat the chambers were placed on cylindrical PVC collars with a height of 25 cm. Collars had
been inserted into the mat to depths of approximately 15 cm prior to the first measurement.
Each sampling day fluxes were measured at least once with the transparent and with the
radiation-shielded chamber, for 5 min on the pond and 3 min on the floating mat, by placing
the chamber gently as soon as the concentration reading was stable. When $CH_4$ concentrations
increased sharply within the first 60 s of the measurement due to $CH_4$ bubble release caused by
the positioning of the chamber, the measurement was discarded and repeated. Fluxes were also
quantified at a peatland site in the north-northeast of the pond (Fig. 1) with the same approach,
every other week from July 4[th] until October 1[st], 2014, on 12 measuring plots covering
hummocks, hollows and lawns. In this area of the peatland, hummocks cover 90 % of the area,
hollows 9.8 % and lawns 0.2 % of the area.
Fluxes were calculated based on the gas concentration change in the chamber over time using
linear regression and the ideal gas law, mean air temperature inside the chamber and the
corresponding half hour mean air pressure. The chamber volume was calculated for each
measurement depending on the number of ice packs, immersion depth on the pond and mean
vegetation height on the floating mat. The first 40 s after chamber deployment were discarded



for flux calculation due to the response time of the concentration measurement. If the slope was
not significantly different from 0 (F test, $\alpha = 0.05$), the flux was set to 0. Concentration change
over time was only $< 3$ ppm $CO_2$ and $< 0.1$ ppm $CH_4$ in 12 % of flux measurements. These
measurements resulted in fluxes close to 0 with $R^2 < 0.8$. Following Repo et al. (2007), we
included them in the data set because their exclusion would have biased the results by increasing
the median diffusive fluxes by 52 % ($CO_2$) and 12 % ($CH_4$).
Due to the high temporal resolution of concentration measurements, we were able to quantify
$CH_4$ fluxes with and without bubbles. When the $CH_4$ concentrations evolved linearly with a
constant slope we used linear regression over the entire time of sampling; when the initial
concentration trend was interrupted by one or several sharp increases in slope, followed by a
return to the initial slope (Supplementary Information, Fig. S1), we used piecewise linear fitting
for each of the linear segments (Goodrich et al., 2011). According to Goodrich et al. (2011) and
Xiao et al. (2014), we define sharp increases in slope as ebullitive $CH_4$ fluxes and all others as
diffusive or continuous flux of micro-bubbles. Time-weighted averages including diffusive and
ebullitive flux segments were calculated. We also computed the $CH_4$ bubble frequency in events
$m^{-2}\,h^{-1}$ as the number of bubble events divided by measuring time and area. In order to evaluate
the contribution of ebullitive $CH_4$ flux to the total $CH_4$ flux, the $CH_4$ release of each event in
µmol was calculated by multiplying the ebullitive flux with the duration of the event and the
basal area of the chamber.
For comparisons of NEE between sites and with time, we used the maximum NEE defined as
light-saturated at PAR levels $> 1000$ µmol $m^{-2}\,s^{-1}$ according to a study by Larmola et al. (2013).
We further calculated the net exchange of $CO_2$ equivalents for each flux measurement. To this
end, the $CH_4$ flux was converted into $CO_2$ equivalents by multiplying the mass flux with the
global warming potential of 28 for a 100 year time horizon (Myhre et al., 2013). Subsequently,
the $CH_4$ flux in $CO_2$ equivalents and the maximum NEE were summed up.
**2.4 $CO_2$ concentration measurements and gradient flux calculations**
Concentrations of $CO_2$ in the surface water of the pond and in the air were measured with
calibrated non-dispersive infrared absorption sensors (CARBOCAB, GMP222, Vaisala) in the
range up to 10000 ppm and with an accuracy of ± 150 ppm plus 2 % of the reading. The probe
was enclosed in $CO_2$ permeable silicone tubes, as already used by Estop et al. (2012) in peats,
and attached to a floating platform at a depth of approximately 18 cm and a distance of 3.2 m
from the pond margin. In water equilibration time to 90% of dissolved concentration was
approximately one hour when concentration increased but more delayed when it fell
(Supplementary Information, Figure S3). The platform also carried the data logger (MI70,
Vaisala). Another silicon-covered sensor measured air $CO_2$ concentrations at 0.3 m above the



water surface. Concentration was recorded every 15 min and $CO_2$ flux across the air-water
interface estimated according to the boundary layer equation approach (Supplementary
Information). Due to frequent failures of the sensors with increased humidity in the sensor head
and overheating of the data logger, $CO_2$ fluxes were only calculated for 5 and 3 exemplary days
in July and September, respectively. During these periods sensor functioning was stable.
**2.5 $CH_4$ and $CO_2$ concentrations and diffusive fluxes in the sediment**
Dissolved $CH_4$ and $CO_2$ concentrations at the sediment-water-interface were determined with
pore water peepers of 60 cm length and 1 cm resolution as developed by Hesslein (1976). The
chambers were filled with deionized water, covered with a nylon membrane of 0.2 μm pore
size, installed at four locations in the pond on August 21st, 2014 and sampled on September 25th
and 29th, 2014. The pH of every other cell was measured in the field and a sample of 0.5 mL
from each chamber filled into a vial containing 20 μL of 4 M hydrochloric acid (HCl). $CO_2$ and
$CH_4$ concentrations in the headspace of the vials were determined with an SRI 8610C gas
chromatograph equipped with a methanizer and a flame ionization detector on the day after
sampling. The original $CO_2$ and $CH_4$ concentrations in the pore water were calculated by using
the measured headspace concentrations, Henry's law with temperature corrected Henry's law
constants (Sander, 1999) and the ideal gas law. Diffusive fluxes of $CO_2$ and $CH_4$ towards the
sediment-water interface were calculated with Fick's first law and diffusion coefficients in
water $D_w$ corrected for an assumed sediment temperature of 15°C ($CH_4$: $1.67 \bullet 10^{-5}$ cm$^2$ s$^{-1}$;
$CO_2$: $1.87 \bullet 10^{-5}$ cm$^2$ s$^{-1}$) and assuming a porosity n of 0.9. The effect of porosity on the sediment
diffusion coefficient was accounted for by multiplying $D_w$ with a factor $n^2$ (Lerman, 1978). We
further calculated a theoretical temperature- and depth-dependent threshold of bubble formation
using Henry's law, correcting Henry's law constant for a temperature of 15°C, and assuming a
partial pressure of $N_2$ in the pore water of 0.8 atm or 0.5 atm. The assumption here is that bubble
formation is possible when the partial pressure of $CH_4$ and remaining $N_2$ exceeds atmospheric
and water pressure in the anoxic sediment.
**2.6 Statistical analyses**
Statistical analyses were performed with R, version 3.1.2 (R Core Team, 2014). All datasets
were checked for normality with the Shapiro-Wilk normality test at a confidence level of
$\alpha = 0.05$. To investigate statistical differences of a continuous variable between two or more
groups, we used the non-parametric Kruskal-Wallis rank sum test ($\alpha = 0.05$) and if applicable
afterwards the multiple comparison test after Kruskal-Wallis ($\alpha = 0.05$) since none of the
datasets were normally distributed. For the investigation of relationships between two
continuous variables, we used Spearman's rank correlation ($\alpha = 0.05$). Due to visually different





dynamics of the gas fluxes from July 10[th] to August 7[th] (here called "mid summer") compared
to August 15[th] to September 29[th] (here called "late summer"), correlations with environmental
variables were examined for the whole period as well as the two subperiods.
**3    Results**
**3.1 Weather conditions**
Three distinct periods of weather occurred. From July 10[th] until September 10[th], 2014, air
temperatures remained high with a mean (± standard deviation) of $17.0 \pm 2.7$ °C (Fig. 2). Most
days were sunny with some passing clouds. From September 11[th] to September 22[nd], 2014,
mean air temperature had cooled to $10.2 \pm 2.8$ °C and the first frost occurred on September 14[th]
(Fig. 2). From September 23[rd] to 29[th], mean air temperature was $13.2 \pm 7.6$ °C with a high daily
amplitude from $3.7 \pm 1.3$ °C to $24.3 \pm 1.5$ °C and wind speed was low with a mean of $0.14 \pm$
$0.31$ m s$^{-1}$ (5 min averages) (Fig. 2). Major storms with maximum wind speeds from 3 to 5.5
m s$^{-1}$ on July 23[rd], July 28[th], August 12[th], September 6[th], September 11[th] and September 21[st]
were accompanied by air pressure decline to minima between 944 and 955 hPa. Often rainfall
reached an intensity of 2.8 to 6.2 mm in the chosen 5 min time intervals (Fig. 2).
**3.2 CH$_4$ and CO$_2$ fluxes over time**
CH$_4$ fluxes from the pond were significantly lower in the period from July 10[th] until August 7[th]
with a median of 0.03 mmol m$^{-2}$ h$^{-1}$ compared to a median of 0.21 mmol m$^{-2}$ h$^{-1}$ from August
15[th] until September 29[th] (Kruskal-Wallis test, $p < 0.001$, n = 159) (Fig. 3 A). The highest
median CH$_4$ flux, highest maximum flux, and largest variability were observed on August 15[th].
The bubble frequency varied between 0 and 30 events m$^{-2}$ h$^{-1}$ (Fig. 3 B) and the contribution
of the ebullitive to the total CH$_4$ flux between 90 % on July 22[nd] and 0 % on September 17[th],
25[th] and 29[th] (Fig. 3 C). Efflux of CH$_4$ from the floating mat was variable but significantly
higher from August 15[th] to September 29[th] with a median of 0.80 mmol m$^{-2}$ h$^{-1}$ than in the
period from July 10[th] to August 7[th] with a median of 0.22 mmol m$^{-2}$ h$^{-1}$ (Kruskal-Wallis test,
$p < 0.001$, n = 84) (Fig. 4 A). The bubble frequency on the floating mat ranged from 0 to 80
events m$^{-2}$ h$^{-1}$ and the contribution of ebullition to CH$_4$ flux from 0 to 88 % (Fig. 4 B and C).
At the peatland site, CH$_4$ fluxes were similar over time with a median of 0.31 mmol m$^{-2}$ h$^{-1}$
and two very high individual fluxes on September 1[st] and October 1[st] (Fig. 5 A). The bubble
frequency and contribution of ebullition to CH$_4$ flux ranged from 0 to 5 events m$^{-2}$ h$^{-1}$ and 0 to
54 %, respectively (Fig. 5 B and C).



$CO_2$ fluxes from the pond from July 10th until August 7th had a median of 0.11 mmol m$^{-2}$ h$^{-1}$
and were also significantly lower than the pond $CO_2$ fluxes from August 15th until September
29th with a median of 1.80 mmol m$^{-2}$ h$^{-1}$ (Kruskal-Wallis test, $p < 0.001$, n = 159) (Fig. 3 D).
During 24 out of 55 individual measurements before August 15th, $CO_2$ exchange across the
water-atmosphere interface was absent or $CO_2$ was taken up by the pond between 0 and
−0.75 mmol m$^{-2}$ h$^{-1}$. From August 15th on $CO_2$ was net emitted. The median daytime ER of the
floating mat was 6.77 mmol m$^{-2}$ h$^{-1}$ and the median of the maximum NEE −4.81 mmol m$^{-2}$ h$^{-1}$
(Fig. 4 D). Daytime ER at the peatland site varied between 2.61 to 36.93 mmol m$^{-2}$ h$^{-1}$ with a
median of 11.98 mmol m$^{-2}$ h$^{-1}$ and tended to decrease towards fall (Fig. 5 D). The maximum
NEE was quite constant from July until September with a median of −16.98 mmol m$^{-2}$ h$^{-1}$.
The gradient method provided similar $CO_2$ fluxes in July and September with a median of
1.99 mmol m$^{-2}$ h$^{-1}$ in July and 2.02 mmol m$^{-2}$ h$^{-1}$ in September (Supplementary Information,
Fig. S2). The daily amplitude of fluxes was 1.46 to 3.19 mmol m$^{-2}$ h$^{-1}$ in July and 1.41 to
1.86 mmol m$^{-2}$ h$^{-1}$ in September (Supplementary Information Fig. S2). In July, however, the
daytime $CO_2$ fluxes obtained by the gradient method were 14-fold higher than the respective
$CO_2$ fluxes measured with the floating chambers (Kruskal-Wallis test, $p < 0.001$, n = 189). No
significant differences occurred between the methods in September.
**3.3 $CO_2$ and $CH_4$ concentrations and diffusion in the surface water and sediments**
$CO_2$ concentrations of the surface water of the pond were similar during the examined periods
in July and September with a mean (± standard deviation) of 114.8 ± 33.1 μmol L$^{-1}$ and 132.0
± 21.0 μmol L$^{-1}$, respectively (Fig. S2, Supplementary Information). In both periods we
observed diurnal cycles of $CO_2$ concentrations covering a mean amplitude of 83.5 ± 16.3
μmol L$^{-1}$ (July) and 62.0 ± 3.1 μmol L$^{-1}$ (September). In the sediments, the mean pH was 4.29
± 0.11 above the sediment-water interface and increased to 5.37 ± 0.28 at a sediment depth of
40 to 60 cm. $CH_4$ concentrations rose with depth from an average of 10.7 ± 20.4 μmol L$^{-1}$ above
the sediment-water interface to 557.3 ± 72.9 μmol L$^{-1}$ at a depth of 40 to 60 cm into the
sediment (Fig. 6). The concentration began exceeding theoretical thresholds for bubble
formation at depths between 10 to 40 cm and at a partial pressure of $N_2$ of 0.8 atm, but nowhere
were concentrations sufficient to form bubbles at 0.5 atm $N_2$ (Fig. 6). The average $CO_2$
concentration at 40 to 60 cm depth was 1548.2 ± 332.5 μmol L$^{-1}$ and one order of magnitude
higher than above the sediment-water interface (Fig. 6). Diffusive fluxes towards the surface
water were on average 10.5 ± 5.6 μmol m$^{-2}$ h$^{-1}$ ($CH_4$) and 16.9 ± 9.4 μmol m$^{-2}$ h$^{-1}$ ($CO_2$), or 12.0
± 5.6 μmol m$^{-2}$ h$^{-1}$ ($CH_4$) and 25.8 ± 16.1 μmol m$^{-2}$ h$^{-1}$, depending on where the concentration
gradient of pore water peeper C is assigned (Fig. 6). *In situ* production and diffusion from the
sediment thus contributed only a very small fraction to the $CO_2$ and $CH_4$ flux from the pond.





1. The relative inactivity of the pond sediment was also indicated by the mostly flat and linear
2. concentration increase of both gases with depth near the sediment-water interface.
3.
4. **3.4 Spatial pattern of $CH_4$ and $CO_2$ fluxes**
5. Efflux of $CH_4$ increased 6-fold from open water towards the floating mat and was also much
6. higher on the floating mat than at the peatland site (Fig. 7 A). The open water median $CH_4$ flux
7. of plot p1, p2 and p3, farthest away from the floating mat, was 0.12 mmol m$^{-2}$ h$^{-1}$ and
8. significantly lower than from plot p4, p5 and p6 closer to the floating mat with a median of 0.19
9. mmol m$^{-2}$ h$^{-1}$ (Kruskal-Wallis test, $p < 0.05$, $n = 82$) (Supplementary Information, Table S3).
10. The median $CH_4$ flux of the floating mat was 0.64 mmol m$^{-2}$ h$^{-1}$ and significantly higher than
11. the $CH_4$ flux from the pond (Kruskal-Wallis test, $p < 0.001$, $n = 243$). We observed an increasing
12. frequency of ebullition and a higher contribution to $CH_4$ flux towards the floating mat. On plot
13. p1 only 4 events m$^{-2}$ h$^{-1}$ contributing 5 % occurred, whereas on plot m3 on the floating mat
14. 168 events m$^{-2}$ h$^{-1}$ contributing 78 % were found (Fig. 7 B and C). The $CH_4$ flux of m3 was
15. significantly higher than of m1 and m2 (Kruskal-Wallis multiple comparison test, $p < 0.05$,
16. $n = 84$).
17. The frequency of ebullition and the amount of $CH_4$ released by bubble events differed along
18. the transect and in comparison to the peatland site. On the pond, bubble events with a
19. comparatively small $CH_4$ release of 0 to 2.5 µmol were most frequent and occurred 5.4 times
20. m$^{-2}$ h$^{-1}$ (Fig. 8). They also contributed the most to the total $CH_4$ release. Bubble events releasing
21. a larger amount of $CH_4$ were rare. The contribution of ebullition to $CH_4$ release was 27 %. On
22. the floating mat, $CH_4$ release by individual bubble events was highly variable with a maximum
23. of 50 µmol (Fig. 8). Larger bubble events were less frequent than smaller ones. However,
24. medium and larger bubble events contributed most to $CH_4$ release with up to 8 %. The
25. contribution of ebullition to $CH_4$ release was 66 % on the floating mat. In contrast, it was only
26. 20 % in the peatland with a clearly different frequency distribution (Fig. 8). Bubble events
27. occurred over a larger range of release strength than on the pond, but they were less frequent
28. with a total bubble frequency of only 1.3 events m$^{-2}$ h$^{-1}$.
29. The pond was on average also a net source of $CO_2$ with a median $CO_2$ efflux of 1.16
30. mmol m$^{-2}$ h$^{-1}$ (Fig. 7 D). On the floating mat, daytime ER ranged from 0.53 to 13.45 mmol
31. m$^{-2}$ h$^{-1}$ and maximum NEE from $-11.46$ to 0.71 mmol m$^{-2}$ h$^{-1}$ (Fig. 7 D).
32.
33. **3.5 Controls on $CH_4$ and $CO_2$ fluxes**
34. $CH_4$ and $CO_2$ fluxes from the pond and ER on the floating mat were significantly negatively,
35. and maximum NEE on the floating mat positively correlated with air, water and mat
36. temperature (Table 1 and 2). We found more negative NEE values at an increasing PAR on the





floating mat as well as on the pond. Late summer fluxes of $CO_2$ and $CH_4$ across the water-
atmosphere interface were positively correlated with wind speed, whereas the respective mid-
summer fluxes were negatively correlated (Table 1 and 2).
$CH_4$ fluxes from the floating mat and the pond were significantly higher for periods with a
decreasing air pressure trend over the last 24 h than for periods with an increasing air pressure
trend (Kruskal-Wallis test, $p < 0.05$ and $p < 0.01$, n = 111 and n = 61) (Fig. 9).
**3.6 Greenhouse gas exchange of the pond system compared to the surrounding peatland**
During our daytime measurements the pond and the floating mat were most frequently
significant net sources of $CO_2$ equivalents, whereas the peatland was generally a sink of $CO_2$
equivalents (Fig. 10; Kruskal-Wallis multiple comparison test, $p < 0.001$, n = 218). The source
strength of $CO_2$ equivalents was largest on the floating mat with a median of 0.21 g $CO_2$
equivalents $m^{-2}$ $h^{-1}$. While the floating mat and peatland site took up $CO_2$ at PAR > 1000 µmol
$m^{-2}$ $s^{-1}$, the pond emitted $CO_2$ to the atmosphere during 90 % of measurements (see Figs. 3, 4,
5). When both greenhouse gases were emitted, $CH_4$ contributed $59 \pm 20$ % to the total emission
of $CO_2$ equivalents of the pond.
**4      Discussion**
**4.1 Spatial pattern of $CH_4$ and $CO_2$ fluxes along the peatland – pond ecotone**
The peatland and especially the floating mat were summer hot spots of $CH_4$ emissions compared
to a variety of sites in other northern peatlands. Fluxes exceeded most, but not all, emissions
recently reported from similar environments by an order of magnitude (Supplementary
Information, Table S4). On a per-day and mass basis mean fluxes reached 204 and 437 mg
$CH_4$-C $m^{-2}$ $d^{-1}$, which is at the high end of fluxes reported in meta-analyses (Olefeldt et al.,
2013. Average $CH_4$ emissions from the open water were still substantial at 63 mg $CH_4$-C $m^{-2}$
$d^{-1}$, which is about 5 times the flux reported from the multi-year study of Stordalen Mire in
Northern Sweden (Wik et al., 2013). Emissions fell, however, well into the range of fluxes
reported from other peatland ponds (Supplementary Information, Table S4). In contrast, $CO_2$
fluxes were fairly inconspicuous compared to fluxes in similar systems (see Supplementary
Information, Table S5, S6); on a per-day and mass basis mean maximum NEE reached −5.4 g
$CO_2$-C $m^{-2}$ $d^{-1}$ in the bog and −1.27 g $CO_2$-C $m^{-2}$ $d^{-1}$ on the floating mat, and daytime ER 3.91
g $CO_2$-C $m^{-2}$ $d^{-1}$ and 1.85 g $CO_2$-C $m^{-2}$ $d^{-1}$, respectively. The pond on average emitted 0.38 g
$CO_2$-C $m^{-2}$ $d^{-1}$. Both pond and floating mat thus lost more $CO_2$ than they fixed during the day,



which suggests that in both environments additional $CO_2$ was released, for example stemming
from carbon-rich groundwater seeping into the pond.
Part of the surprising source strength of methane can be attributed to the inclusion of ebullition
by means of high frequency chamber measurements, similarly as first reported by Goodrich et
al. (2011). Fluxes that are visibly affected by ebullition events have often been discarded from
static chamber fluxes in the past because the non-linearity of concentration increase over time
is problematic when few samples are analyzed by gas chromatography. Ebullition contributed
on average 66 % to the emissions on the floating mat and reached 78 % at the plot with the
highest methane flux (Figs. 4 and 7). The importance and variability of ebullition was similar
as reported from an ombrotrophic peatland in Japan (50 to 64 %; Tokida et al., 2007). The $CH_4$
released by individual bubble events from the floating mat was also on the same order of
magnitude as bubble $CH_4$ release in Sallie's Fen (Goodrich et al., 2011). At that site the bubble
frequency of $35 \pm 16$ events $m^{-2}$ $h^{-1}$ was, however, lower than on the floating mat at Wylde
Lake Bog with 54 to 168 events $m^{-2}$ $h^{-1}$. In contrast to these findings, ebullition accounted on
average only for 20 % of fluxes at our bog site and 27 % in the pond (Figs. 3 and 5), where
bubble frequency of outer plots was less than 9 events $m^{-2}$ $h^{-1}$ and dropped to zero by the end
September (Fig. 3). In the pond ebullition was thus less important than reported previously in
11 lakes in Wisconsin (40 to 60 %; Bastviken et al., 2004) and two productive, urban ponds in
Sweden and China (> 90 %; Natchimuthu et al., 2014; Xiao et al., 2014).
Even though bubbles were rarely observed on p1, p2 and p3 farther away from the floating mat
(Fig. 7) and ceased altogether in September (Fig. 3), formation of $CH_4$ bubbles may have
initially been possible in the pond sediments. Concentrations exceeded the threshold
concentration of bubble formation at a $N_2$ partial pressure of 0.8 atm in all locations sampled
(Fig. 6). Such concentrations were only reached at larger sediment depth, though, and ongoing
stripping of $N_2$ with ebullition may have raised concentration thresholds over time (Fechner-
Levy and Hemond, 1996). At a remaining $N_2$ partial pressure of 0.5 atm, ebullition was not
possible from a theoretical point of view, which may explain its limited importance in the pond.
The lack of ebullition later on may have been assisted by falling temperatures in autumn; a
change from 20°C to 10°C, for example, raises the threshold for ebullition by 70 $\mu mol$ $L^{-1}$. Flat
or linearly increasing concentration profiles near the sediment-water interface (Fig. 6) also
indicated a lack of active production of the gas in this zone. Concentrations of $CH_4$ and $CO_2$
remained low, typically less than 650 and 1500 $\mu mol$ $L^{-1}$, respectively, suggesting that
microbial activity in the sediments was limited. Also the diffusive fluxes were small in units of
mass, about 3.5 mg $CH_4$-C $m^{-2}$ $d^{-1}$ and 7.5 mg $CO_2$-C $m^{-2}$ $d^{-1}$, respectively. The continuous
emission of $CH_4$ and $CO_2$ from the pond, on average 63 mg $CH_4$-C $m^{-2}$ $d^{-1}$ and 380 mg $CO_2$-C





m$^{-2}$ d$^{-1}$, was hence likely driven by respiration in the water column and by advective inflow of
groundwater rich in $CH_4$ and $CO_2$.
Our results further suggest that medium and infrequent large bubble events contributed a
substantial fraction to the total $CH_4$ flux at the floating mat but not in the bog and the pond (Fig.
8). This was the case even though small bubble events were much more frequent than large
ones (Fig. 8). DelSontro et al. (2015) also reported a strong positive correlation between
ebullition flux and bubble volume in open water and found that the largest 10 % of the bubbles
observed in Lake Wohlen, Switzerland, accounted for 65 % of the $CH_4$ transport. According to
the authors, large bubbles are disproportionately important because they contain exponentially
more $CH_4$ with increasing diameter, rise faster, and have less time and a relatively smaller
surface area to dissolve or exchange $CH_4$ with the surroundings (DelSontro et al., 2015).
The decline of $CH_4$ fluxes, $CH_4$ bubble frequency and contribution of ebullition from the
floating mat to the open water was striking and fluxes were also considerably higher than at the
peatland site (Fig. 7). These findings emphasize that the floating mats and transition zones to
the open water need to be included when quantifying greenhouse gas budgets of pond and
peatland ecosystems. We cannot mechanistically identify the causes for the observed pattern.
It seems likely that the peak emissions from the floating mat were caused by an optimum of
wet conditions in the peat favoring methanogenesis and impeding methane oxidation, presence
of some *Carex aquatilis* providing for conduit transport of the gas (Supplementary Information,
Table S1), and potentially by a release of methane from groundwater entering the land-water
interface. $CH_4$ flux through plants with aerenchymatic tissues can be responsible for 50 to 97 %
of the total $CH_4$ flux in peatlands because the aerenchyma link the anaerobic zone of $CH_4$
production with the atmosphere (Kelker and Chanton, 1997; Kutzbach et al., 2004; Shannon et
al., 1996). Kutzbach et al. (2004) found a strong positive correlation between the density of *C.*
*aquatilis* culms and $CH_4$ fluxes, as well as a contribution of 66 ± 20 % of the plant-mediated
$CH_4$ flux through *C. aquatilis* to the total flux in wet polygonal tundra. Since ebullition
dominated the $CH_4$ flux from the floating mat (Fig. 4) in our particular case this transport
mechanism seemed to be of more limited importance, though. Also recently fixed substrates
may have played a role for high $CH_4$ emissions from the floating mat. Several studies have
found a positive correlation between the rate of photosynthesis and $CH_4$ emissions (Joabsson
and Christensen, 2001; Ström et al., 2003), which has been explained by the quick allocation
of assimilated labile carbon to the roots and subsequent exudation to the anaerobic rhizosphere
(Dorodnikov et al., 2011). These recent photosynthates serve as a preferential source of $CH_4$
compared to older more recalcitrant organic matter (Chanton et al., 1995). Labile organic matter
produced by vascular plants was probably also imported from the floating mat to the margin of
the pond (Repo et al., 2007; Wik et al., 2013). Given the gradual decline of $CH_4$ fluxes along





the transect CH$_4$-rich groundwater may also have entered the floating mat and the pond, a
process that we did not investigate.
**4.2 Controls on CH$_4$ and CO$_2$ fluxes**
In agreement with earlier work air pressure change influenced methane flux. We observed 1.5-
to 3-fold higher CH$_4$ fluxes from the floating mat and the pond during periods of decreasing
compared to increasing air pressure (Fig. 9), which was very likely caused by increased
ebullition (Wik et al., 2013). Decreased atmospheric pressure results in bubble expansion,
which enhances buoyancy force and entails bubble rise (Chen and Slater, 2015).
The negative correlation of water and mat temperature with CH$_4$ and CO$_2$ fluxes from the pond
and CH$_4$ flux and ER of the floating mat (Table 1 and 2) was unexpected, as it is consensus that
temperature is an important positive control on these fluxes (Pelletier et al., 2014; Roulet et al.,
1997; Sachs et al., 2010; Wik et al., 2014). Also the potential effect of wind speed on CH$_4$ and
CO$_2$ fluxes from the pond was ambiguous. Increasing wind speeds should stimulate the
exchange of dissolved gases by increasing turbulence of both air and water close to the interface
(Crusius and Wanninkhof, 2003). Before August 15$^{th}$, wind speed and CH$_4$ and CO$_2$ efflux from
the pond were, however, negatively correlated, whereas the correlation was positive thereafter
despite quite consistent wind speed patterns and surface water CO$_2$ concentrations throughout
the whole study period (Figs. 2 and S2, Supplementary Information).
Both phenomena may be explained by internal biological processes, i.e. the growth and decay
of a dense algal mat on the pond, changing hydrological connection between the pond system
and the surrounding peatland, and the influence of the vascular vegetation on the floating mat.
The algal mat developed in the beginning of July and was largely irreversibly dissolved by a
storm on August 12$^{th}$ (Figs. S4 and S5). During its presence CO$_2$ emissions from the pond
remained low (Fig. 3) and were overestimated by the boundary layer equation approach.
Amplitudes of dissolved CO$_2$ concentration were strong and concentration decreased with
increasing PAR (Table 1). Such dynamics reflects a strong autochthonous photosynthetic and
respiratory activity and lack of water mixing. The empirical relationship between CO$_2$
concentration gradient, wind speed and flux, which is largely controlled by turbulence in the
water column, obviously did not apply under such conditions. The subsequent shift to high CO$_2$
and CH$_4$ emissions was probably partly caused by the decomposition of the remains of the algal
mat, similarly as reported from a boreal and a subtropical pond (Hamilton et al., 1994; Xiao et
al., 2014). Other than that, the algal mat probably represented a physical barrier to diffusive and
ebullitive gas exchange between water column and atmosphere. We observed trapped gas
bubbles within the algal mat with CH$_4$ concentration of only 4 to 8 %; part of the originally
contained CH$_4$ may have been re-dissolved and oxidized. Even in shallow lakes and ponds, CO$_2$



and $CH_4$ concentrations can be several-fold higher in the deep water compared to the surface
water during certain periods (Dinsmore et al., 2009; Ford et al., 2002). We can only assume
that such concentration gradients established in or under the algal mat. Its destruction, mixing
of the water column and resuspension of the upper sediment layer probably entailed the
observed peak diffusive $CO_2$ and $CH_4$ emissions after the storm on August 12th (Fig. 2, Fig. 3).
**4.3 Relevance of greenhouse gas emissions from the pond system**
In terms of radiative forcing, the floating mat and open water behaved differently than the
peatland site during our daytime flux measurements at PAR > 1000 µmol $m^{-2}\,s^{-1}$. All three bog
micro-sites represented daytime sinks of $CO_2$ equivalents and most so the hummocks (Fig. 10),
which represented about 90 % of the area. The floating mat and to a lesser extent also the pond
were sources of $CO_2$ equivalents to the atmosphere, even at daytime, and had a comparable
source strength as the boreal ponds and beaver pond investigated by Hamilton et al. (1994) and
Roulet et al. (1997). Net photosynthetic $CO_2$ uptake at light saturation was thus unable to
counterbalance the high $CH_4$ emissions of the floating mat in terms of $CO_2$ equivalents; at both
the floating mat and the pond emission of $CH_4$ was more important than $CO_2$ exchange in terms
of greenhouse gas equivalents. In the pond the average contribution of $CH_4$ was 59 %, which is
much higher than reported from a beaver pond at the Mer Bleue bog (5 %; Dinsmore et al.,
2009), but comparable to figures from ponds in other studies (36 to 91 %; Hamilton et al., 1994;
Huttunen et al., 2002; Pelletier et al., 2014; Repo et al., 2007; Roulet et al., 1997). We ascribe
the large differences between the floating mat and the peatland site (Figs. 7 and 10) to the influx
of allochthonous organic and inorganic carbon to the pond system from the surroundings and
to the different vegetation composition, in particular the occurrence of *Carex aquatilis* on the
floating mat, which may have enhanced $CH_4$ production and transport (Kutzbach et al., 2004;
Strack et al., 2006). Our results support earlier suggestions that ponds are important for the
greenhouse gas budget of peatlands at landscape scale (e.g. Pelletier et al. 2014) and they
suggest that changes in the area extent of floating mats and shore length will be an important
factor of changes in greenhouse gas budgets with predicted climate change.
**5      Conclusions**
Our summer measurements of atmospheric $CH_4$ and $CO_2$ exchange revealed a substantial small-
scale spatial variability with 6- and 42-fold increasing median $CH_4$ fluxes and bubble
frequencies, respectively, from the open water of the pond towards the surrounding floating
mat. Individual bubble events releasing more than 10 µmol $CH_4$ contributed substantially to
summer $CH_4$ emissions from the floating mat, despite their rare occurrence. When $CH_4$



emissions of peatlands that contain ponds are quantified, ebullitive and diffusive $CH_4$ fluxes at
the land-water interface hence need to be accounted for and the areal cover of the different
microforms and/or plant communities should be thoroughly mapped, as suggested by Sachs et
al. (2010) for tundra landscape. We also observed 4- to 16-fold increases in $CH_4$ and $CO_2$
emissions in late summer that were unrelated to meteorological drivers, such as temperature,
wind speed and radiation. Hydrological connections to adjacent peatlands and internal
hydrological and biological processes, such as the development of algal mats, which can be
abundant in small and shallow water bodies (e.g. Dinsmore et al., 2009; Hamilton et al., 1994;
Xiao et al., 2014) thus require more attention in the future. During our summer daytime flux
measurements, the pond system had a warming effect considering $CH_4$ and $CO_2$ exchange, with
the highest net release of $CO_2$ equivalents from the floating mat. We conclude that carbon
cycling and hydrology of small ponds and their surrounding ecotone need to be further
investigated; these systems are hot spots of greenhouse gas exchange and are likely highly
sensitive to anthropogenic climate change due to their shallowness and dependence on water
budgets and hydrological processes upstream.
**Acknowledgements**
The study was financially supported by the German Research Foundation (DFG) grant BL
563/21-1 and an international cooperation grant by the German Academic Exchange Service
(DAAD) to C. Blodau. We thank C. Wagner-Riddle for the possibility to use the former Blodau
laboratory at the School of Environmental Sciences at the University of Guelph and P. Smith
and L. Wing for organizational and technical support. We are grateful to M. Neumann from the
Grand River Conservation Authority for permission to conduct research in the Luther Marsh
Wildlife Management Area, Ontario, Canada, Z. Green for kindly providing satellite images of
the study area and C.A. Lacroix (OAC Herbarium, Biodiversity Institute of Ontario) for her
friendly help in identifying some plants. We are thankful to M. Goebel for support in the field
and advices on study design and data analysis and to Elisa Fleischer for her helpful comments.



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





Table 1. Correlations of $CH_4$ and $CO_2$ fluxes of the pond with environmental variables.

| Flux | Time period | Spearman's rho | p | n |
|------|-------------|----------------|---|---|
| *mean air temperature since sunrise* | | | | |
| $CO_2$ | whole period | − 0.54 | < 0.001 | 147 |
| $CH_4$ | whole period | − 0.36 | < 0.001 | 147 |
| diffusive $CH_4$[a] | whole period | − 0.67 | < 0.001 | 119 |
| *mean water temperature during measurements* | | | | |
| $CO_2$ | whole period | − 0.47 | < 0.001 | 94 |
| $CH_4$ | whole period | − 0.50 | < 0.001 | 94 |
| diffusive $CH_4$[a] | whole period | − 0.60 | < 0.001 | 82 |
| *mean PAR of the last 3 h* | | | | |
| $CO_2$ | whole period | − 0.49 | < 0.001 | 147 |
| *mean wind speed of the last 24 h* | | | | |
| $CO_2$ | mid summer[b] | − 0.35 | < 0.05 | 43 |
| $CO_2$ | late summer[c] | + 0.45 | < 0.001 | 104 |
| $CO_2$ | whole period | not significant | | |
| $CH_4$ | mid summer[b] | − 0.35 | < 0.05 | 43 |
| $CH_4$ | late summer[c] | + 0.63 | < 0.001 | 104 |
| $CH_4$ | whole period | + 0.26 | < 0.01 | 147 |
| *maximum wind speed of the last 24 h* | | | | |
| $CO_2$ | mid summer[b] | − 0.45 | < 0.01 | 43 |
| $CO_2$ | late summer[c] | + 0.35 | < 0.001 | 104 |
| $CO_2$ | whole period | + 0.17 | < 0.05 | 147 |
| $CH_4$ | mid summer[b] | − 0.55 | < 0.001 | 43 |
| $CH_4$ | late summer[c] | + 0.63 | < 0.001 | 104 |
| $CH_4$ | whole period | + 0.32 | < 0.001 | 147 |

[a]: only measurements without ebullition included
[b]: July 10th to August 7th
[c]: August 15th to September 29th



Table 2. Correlations of $CH_4$ and $CO_2$ fluxes of the floating mat with environmental variables.

| Flux | Time period | Spearman's rho | p | n |
|---|---|---|---|---|
| *mean air temperature since sunrise* | | | | |
| max. NEE | whole period | + 0.74 | < 0.001 | 20 |
| $CH_4$ | whole period | − 0.42 | < 0.001 | 79 |
| *mean mat temperature during measurements* | | | | |
| ER | whole period | − 0.44 | < 0.01 | 38 |
| $CH_4$ | whole period | − 0.41 | < 0.001 | 79 |
| diffusive $CH_4$[a] | whole period | − 0.52 | < 0.001 | 53 |
| *mean PAR during measurements* | | | | |
| NEE | mid summer[b] | not significant | | |
| NEE | late summer[c] | − 0.60 | < 0.01 | 26 |
| NEE | whole period | − 0.37 | < 0.05 | 42 |

[a]: only measurements without ebullition included
[b]: July 10th to August 7th
[c]: August 15th to September 29th





**Figures**

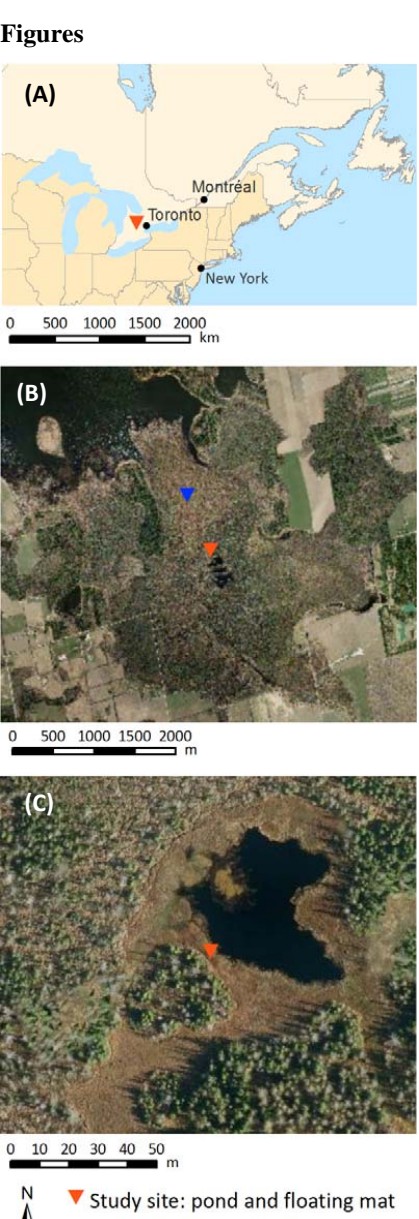

Figure 1. Location of the study site in southern Ontario, Canada (panel A), studied pond with floating mat and peatland site in Wylde Lake Bog in the Luther Marsh Wildlife Management Area with Luther Lake in the northwest (panel B) and close-up of the studied pond and floating mat (panel C) (Grand River Conservation Authority, 2010)



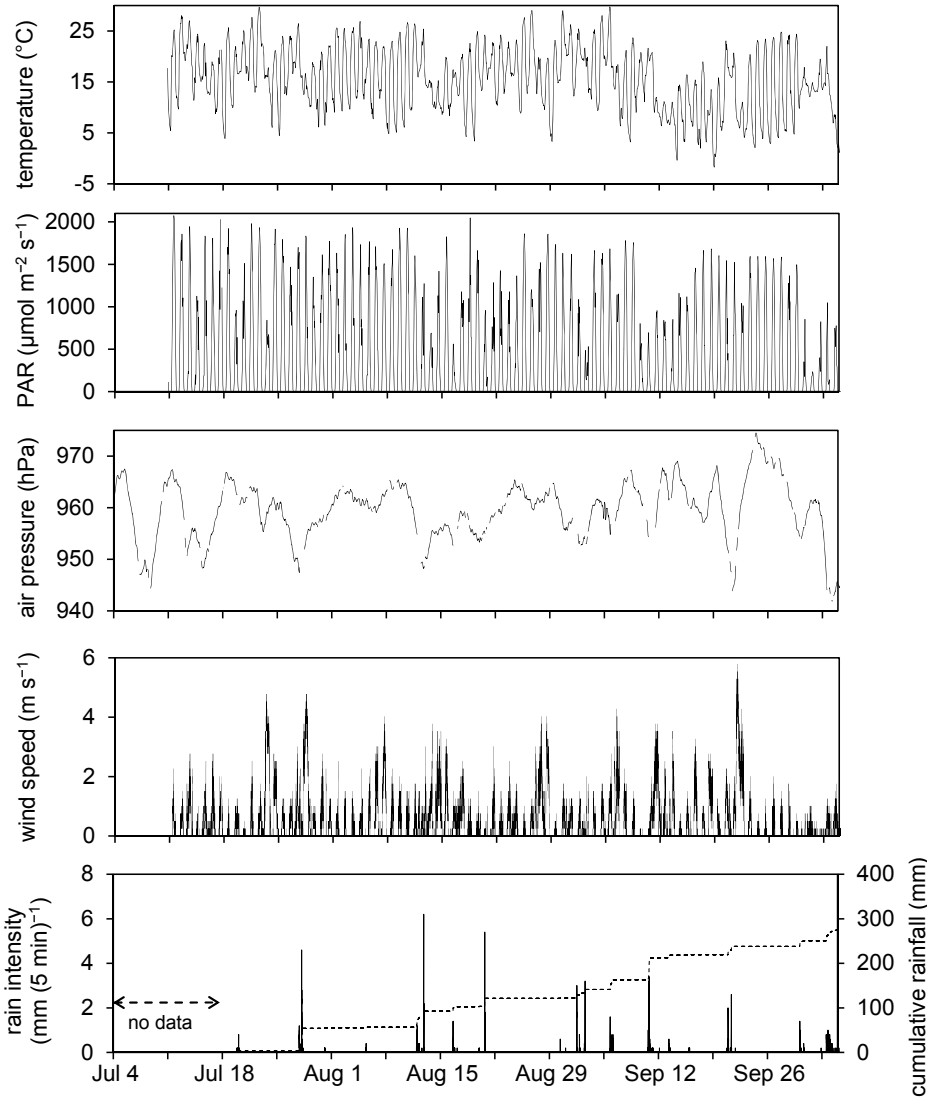

Figure 2. Time series of weather variables at the study site. Air temperature, photosynthetically active radiation (PAR) and air pressure are shown as hourly means, wind speed and rain intensity as 5 min averages. The dashed line in the lowest panel shows the cumulative rainfall.





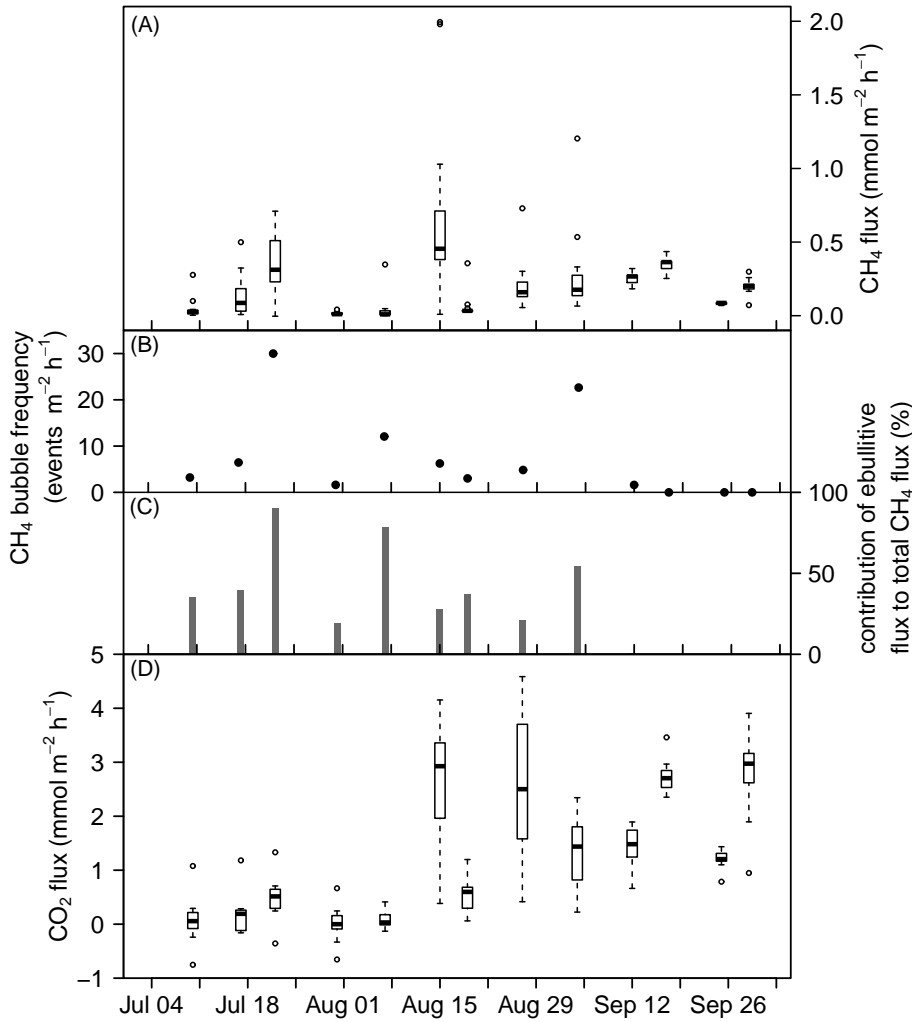

Figure 3. Time series of pond $CH_4$ fluxes (panel A), $CH_4$ bubble frequency (panel B), contribution of ebullitive $CH_4$ flux to total $CH_4$ flux (panel C) and $CO_2$ fluxes (panel D) on measuring days from July 10[th] until September 29[th], 2014. In panel (A) and (D), the bold horizontal line shows the median, the bottom and the top of the box the 25[th] and 75[th] percentile and the whiskers include all values within 1.5 times the interquartile range.





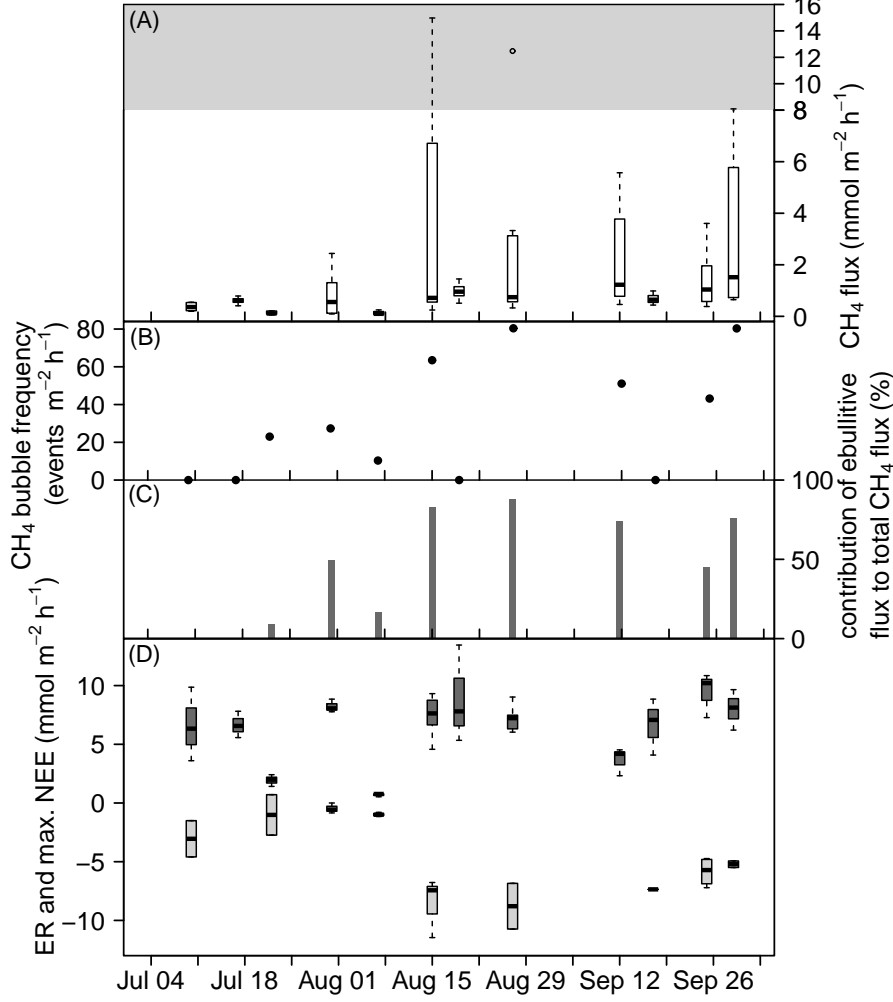

Figure 4. Time series of floating mat $CH_4$ fluxes (panel A), $CH_4$ bubble frequency (panel B), contribution of ebullitive $CH_4$ flux to total $CH_4$ flux (panel C) as well as ecosystem respiration (ER) and maximum net ecosystem exchange (NEE) (panel D) on measuring days from July 10[th] until September 29[th], 2014. Note the different scaling of the y-axis within the gray area in panel (A). In panel (D), the dark gray boxes show the daytime ER and the light gray boxes the maximum net ecosystem exchange at values of photosynthetically active radiation > 1000 μmol $m^{-2}$ $s^{-1}$. In panel (A) and (D), the bold horizontal line shows the median, the bottom and the top of the box the 25[th] and 75[th] percentile and the whiskers include all values within 1.5 times the interquartile range.




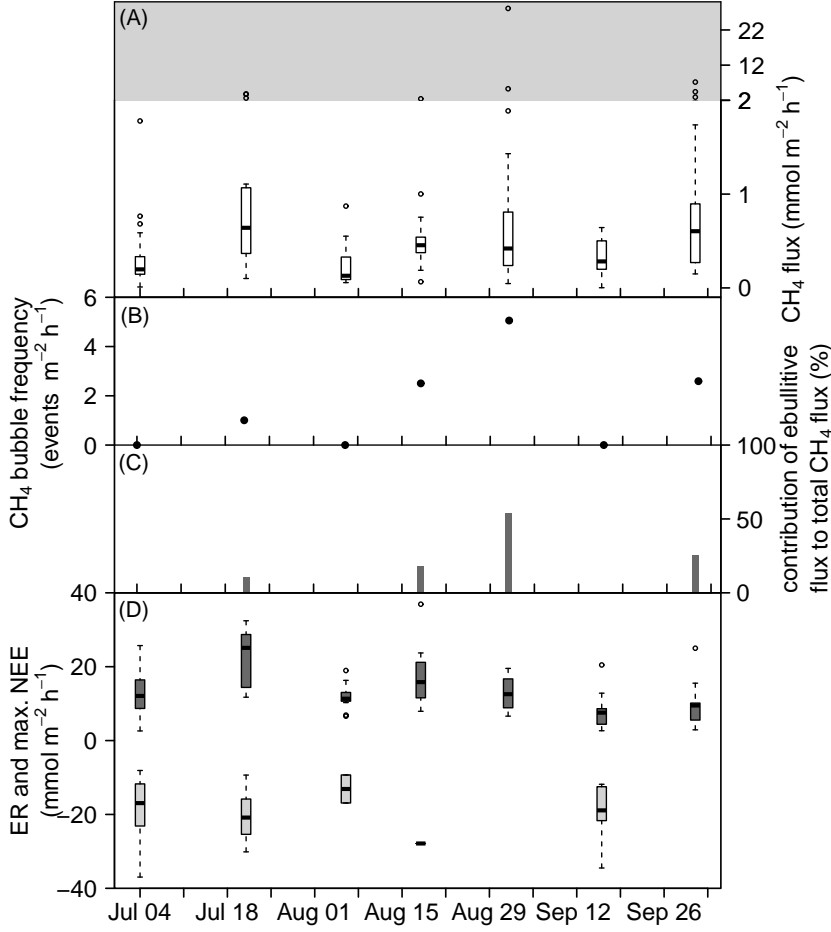

Figure 5: Time series of peatland CH$_4$ fluxes (panel A), CH$_4$ bubble frequency (panel B), contribution of ebullitive CH$_4$ flux to total CH$_4$ flux (panel C) as well as ecosystem respiration (ER) and maximum net ecosystem exchange (NEE) (panel D) on measuring days from July 4$^{th}$ until October 1$^{st}$, 2014. Note the different scaling of the y-axis within the gray area in panel (A). In panel (D), the dark gray boxes show the daytime ER and the light gray boxes the maximum net ecosystem exchange at values of photosynthetically active radiation > 1000 µmol m$^{-2}$ s$^{-1}$. In panel (A) and (D), the bold horizontal line shows the median, the bottom and the top of the box the 25$^{th}$ and 75$^{th}$ percentile and the whiskers include all values within 1.5 times the interquartile range.





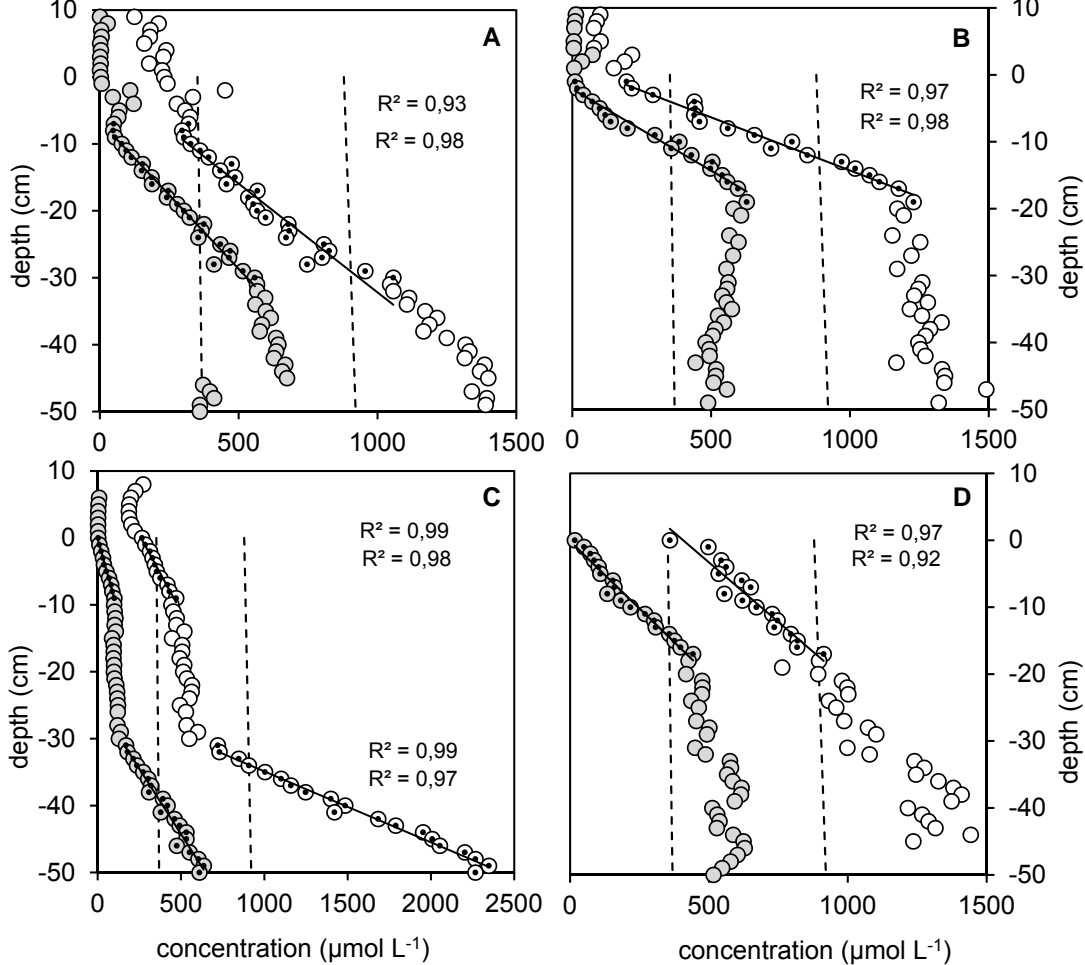

Figure 6: $CH_4$ (shaded symbols) and $CO_2$ (open symbols) concentrations near the sediment-water interface and in the sediment of the pond in four locations (A – D) on September 25[th] and 29[th], respectively. Black lines represent regression slopes (with regression coefficient $R^2$) used to calculate diffusive fluxes towards the sediment-water interface. Dashed lines denote depth and temperature dependent theoretical thresholds for formation of $CH_4$ bubbles at 0.8 atm (lower line) and 0.5 atm (upper line) partial pressure of $N_2$ in the pond sediment at 15°C. In panel C also the diffusive flow from deeper



sediment layers was calculated.

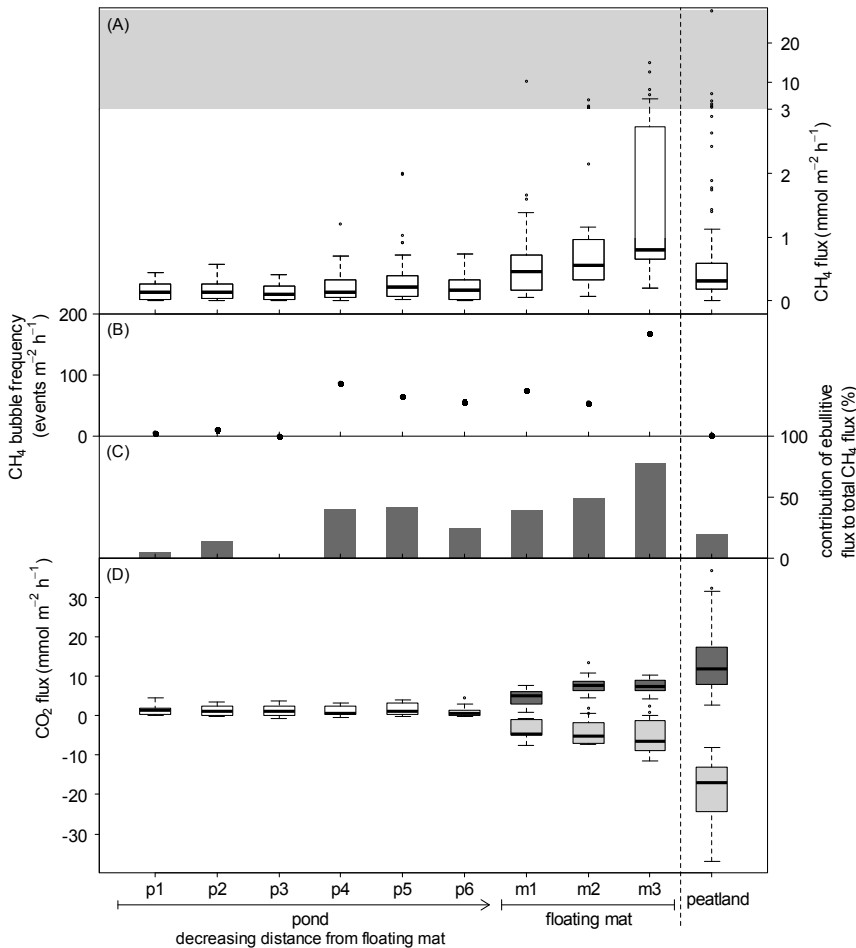

Figure 7: $CH_4$ fluxes (panel A), $CH_4$ bubble frequency (panel B), contribution of ebullitive $CH_4$ flux to total $CH_4$ flux (panel C) and $CO_2$ fluxes (panel D) of the pond (p1 to p6) along a gradient of decreasing distance from the floating mat, of the 3 measuring plots on the floating mat (m1 to m3) and of the peatland site for comparison. Note the different scaling of the y-axis within the gray area in panel (A). In panel (D), the transparent boxes show the net $CO_2$ flux of the pond, the dark gray boxes the daytime ER and the light gray boxes the maximum net ecosystem exchange of the floating mat and the peatland at values of photosynthetically active radiation > 1000 μmol $m^{-2}$ $s^{-1}$. In panel (A) and (D), the bold horizontal line shows the median, the bottom and the top of the box the 25$^{th}$ and 75$^{th}$ percentile and the



whiskers include all values within 1.5 times the interquartile range.

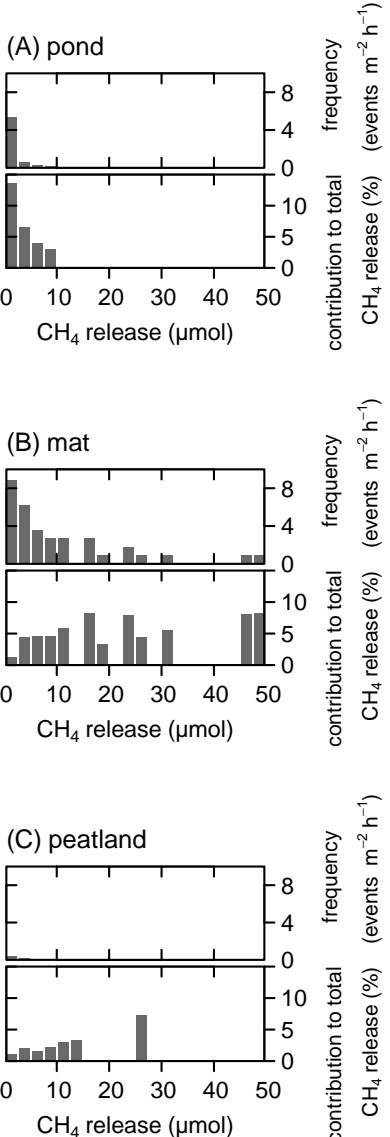

Figure 8: Frequency distribution of ebullitive CH$_4$ release (upper panels) as well as contribution of each size group of ebullitive CH$_4$ release to the total CH$_4$ release (lower panels) of the pond (panel A), the floating mat (panel B) and the peatland (panel C).



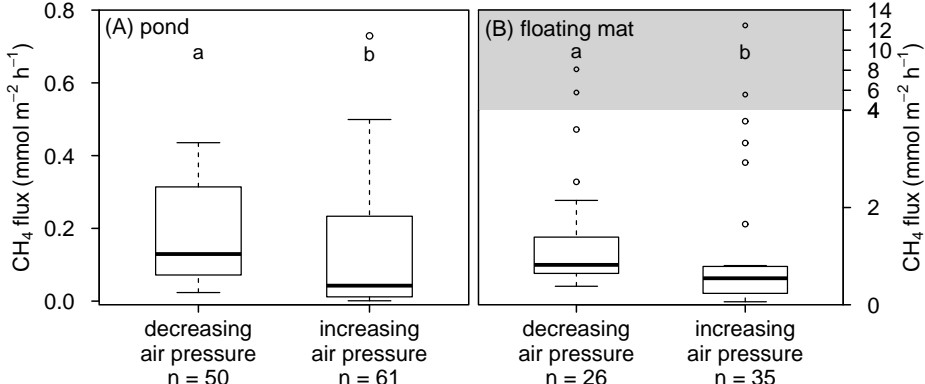

Figure 9: CH$_4$ fluxes during decreasing and an increasing air pressure trends over the last 24 h for the pond (panel A) and the floating mat (panel B). Different letters indicate significant differences (Kruskal-Wallis test, $p < 0.05$ and $p < 0.01$, n = 111 and n = 61). Note the different scaling of the y-axis within the gray area in panel (B). The bold horizontal line shows the median, the bottom and the top of the box the 25$^{th}$ and 75$^{th}$ percentile and the whiskers include all values within 1.5 times the interquartile range.





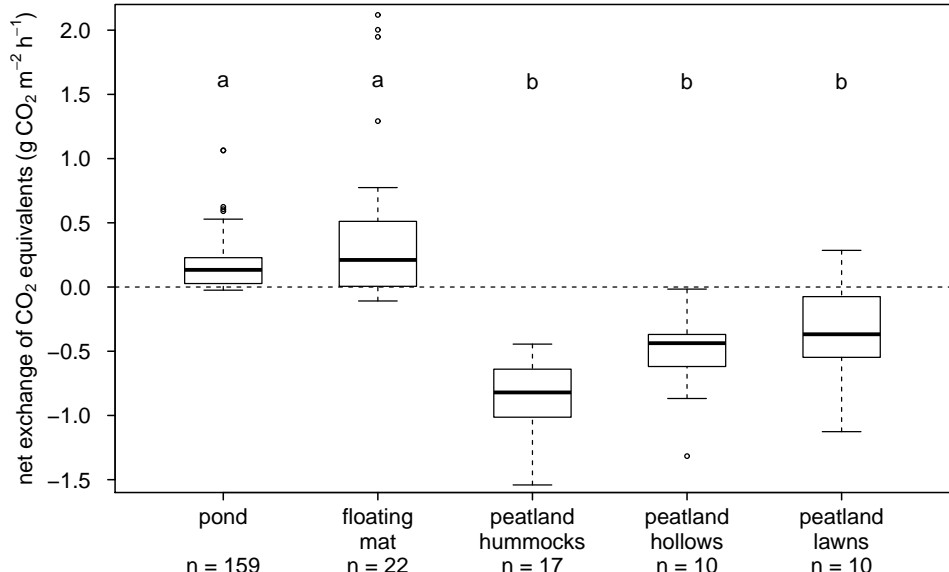

Figure 10: Daytime net exchange of $CO_2$ equivalents of the pond, the floating mat and the three different microforms of the peatland. Different letters indicate significant differences (Kruskal-Wallis multiple comparison test, $p < 0.001$, $n = 218$). For comparability of the $CO_2$ fluxes of the floating mat and the peatland, only maximum net ecosystem exchange at values of photosynthetically active radiation > 1000 µmol m$^{-2}$ s$^{-1}$ was used for the calculation. The bold horizontal line shows the median, the bottom and the top of the box the 25$^{th}$ and 75$^{th}$ percentile and the whiskers include all values within 1.5 times the interquartile range.