# Peer review of "Summer fluxes of methane and carbon dioxide from a pond"

_Biogeosciences, 2015_

## Referee Comment (RC1) · Anonymous Referee #2 · 5 Feb 2016

Review of Burger, Berger, Spangenberg, and Blodau: Summer fluxes of methane and carbon dioxide from a pond

General comments The manuscript reports CH4 and CO2 exchange along a peatland-floating mat-pond transect in southern Ontario, measured in summer period. The scientific basis was clearly defined and the study design was appropriate. The objective of the study was to determine the spatial variation in CH4 and CO2 fluxes in a peatland pond environment and specifically to assess the role peatland ponds and pond-peatland transition as CH4 and CO2 sources. This is a topical issue as many studies have highlighted high process rates in small aquatic habitats, which are numerous, but small in area. Improved spatial resolution of remote sensing techniques may help to

account these in landscape analyzes in near future. Authors set two hypothesis: (I) ebullitive and diffusive CH4 fluxes increase from the open water towards a floating mat surrounding the pond. (II) if CH4 and CO2 effluxes from the system increases with temperature and wind speed, and falling air pressure. These are okay, but maybe authors could use also the background research question as a hypothesis: peatland ponds with the floating edge/transition are significant elements in the peatland C-exchange.

The methodology was valid, measurements conducted properly, results were clearly presented and conclusions were supported by the measurements. The manuscript was generally well written, but many references to supplementary information was irritating to my opinion. I recommend to consider better ways to refer to the supplement and diminish it's role. I found the results section hard to read, because it was loaded with numbers i.e. listing means, medians and errors of the flux rates and results of statistical tests. Overall, there were very many statistical comparisons, and I think that not all of them are necessary to test the hypotheses. The comparison of mid summer and end summer was not motivated and it is not related to a hypotheses. Was the question related to the algal mats? The role of algal mats on the seasonality of C-gas fluxes, mentioned in the discussion, could be an interesting theme if better included in the story.

Specific comments and technical corrections

Abstract, please define CO2 equivalents also in the abstract. p. 1., l. 19. Hot spot of CH4 emissions in summer

p. 2., l. 27. The focus in on peatland ponds here but it is relevant also recognize some lake wide studies showing that the terrestrial –aquatic interface, often vegetated, has high CH4/CO2 fluxes relative to the area. e.g. Juutinen et al. 2003 GBC; Larmola et al. 2004 JGR.

P 3., l. 6, Maybe include some last century papers about the importance of plant mediated transport in ecosystems. There are many.

[Figure]

P. 3, L. 18. You could try to explain this. For example, in Huttunen et al. 2003 (Chemosphere) the deep of a eutrophic lake had the highest ebullition rates, because it is the site of highest sediment accumulation.

p. 3, introduction. Acknowledge the pond work conducted in the Hudson Bay Lowlands.

p. 3, l. 27. Hypothesis II. Rewrite to keep the consistency in the style.

p. 4., l. 27. Is the supplementary information necessary?

p. 5., l. 3-4, 6. Give the sensor make and model in the main text. Please, specify how you arranged the ice packs for the cooling. Did the packages block irradiation inside the chambers?

p. 11, l. 5-6. Total flux? I.e. ebullition and flux summed? Add this information also into fig. 9.

p. 11, l. 8- This is the most interesting piece. Maybe compress the previous results section to make this stronger and report gases as CH4 and CO2 (now only CO2 equivalents). Could you do a spatial extrapolation too? How the pond and floating mat contribute within the whole peatland 'complex'

p. 11 Discussion. Discuss how the features of the floating mat could affect 'physically' to the flux'. How you secured that the measurement/measurer did not cause disturbance increasing bubble release? Is it possible that gases are trapped under the floating mat? Is the water under the mat enriched with the gases? Etc..

p. 11, l. 23-24. Please, include some of the references in the main document. Supplement does not work well in this way.

p. 13, l. 19. Reference to literature instead of Supplementary.

I did not checked the reference list.

———————————————————————

---

## Referee Comment (RC2) · Anonymous Referee #1 · 8 Feb 2016

This manuscript makes an important contribution to the literature on carbon dioxide and methane fluxes from small ponds, which play a significant role carbon-rich and spatially complex peatlands. Throughout the text, the authors do a good job of situating their study within the wider literature on this topic. The chamber methods are robust and well described. However, I am confused about the presentation of the results from the $CO_2$ gradient method. The authors describe that "the gradient method provided similar $CO_2$ fluxes in July and September . . ." but later that, "in July, however, the daytime $CO_2$ fluxes obtained by the gradient method were 14-fold higher than the respective $CO_2$ fluxes measured with the floating chambers". It is unclear to me whether the authors were comparing the gradient $CO_2$ fluxes in both July and September (and

if so, why should they be similar?) or whether they are comparing to the chamber measurements. In my opinion, the authors might add another sentence or two in the methods text to justify why they used the gradient technique, and why fluxes calculated by this method can differ from surface chamber measurements. The authors do detail these mechanisms later on in the discussion, but they might be better motivated if presented earlier in the text.

There are many figures associated with this manuscript, and it sometimes becomes difficult to digest all of the information presented within them. I believe that Figure 9 might be eliminated and that the information within it can be added to the other information in Tables 1 and 2 that detail relationships between environmental drivers and $CO_2$ and $CH_4$ flux. Furthermore, Figure 2 might be added to the Supplementary Material, since it is not thoroughly discussed within the results or discussion. In Figure 6, it's unclear which four locations are depicted within each panel, and this information is not in the Methods section of the main text. It's also unclear where the sediment/water interface is located in Figure 6, and how deep the water is at each location. This information would be valuable for interpreting the discussion surrounding Figure 6 at the end of page 12. Also, I assume that these data from the peepers and not the Vaisala probes, but it would be good to indicate this within the figure caption.

Minor points: Table S1: Reference error in the caption

---

## Author Comment (AC1) · 16 Apr 2016

Reviewer 1:

It is unclear to me whether the authors were comparing the gradient CO2 fluxes in both July and September or whether they are comparing to the chamber measurements.

– Here we compared the results of the gradient method with the chamber measurements at two different times, a) in September when the results were similar, and b) in July when fluxes obtained with the gradient method were overestimated. We suggest clarifying the wording of this statement in the revised version of the manuscript.

The authors might add another sentence or two in the methods section to justify why

they used the gradient technique and why fluxes and why they differ from chamber measurements.

– We used the gradient method coupled to infrared sensor measurements to obtain daily amplitudes of $CO_2$ concentrations and fluxes, which is difficult to do with chamber measurements. We agree with the reviewer and suggest adding this motivation to the methods section.

I believe that Figure 9 might be eliminated and that the information within it can be added in tables 1 and 2.

– As the reviewer rightly states, Tables 1 and 2 detail relationships between environmental drivers and $CO_2$ and $CH_4$ flux. The relationships consistently consist of correlations, however, which differs from the comparison of means with the Kruskal Wallis test. To reduce the number of figures, as requested, we thus suggest adding the four numbers and standard deviations of the fluxes to the text, and moving the figure itself, which essentially visualizes the distribution of flux values, to the Supplementary information.

Figure 2 might be added to the Supplementary Material since it is not thoroughly discussed within the results and discussion.

– In the results we wrote a section on the weather data, which is in our opinion helpful for evaluation of the flux data, in particular with regard to the presentation of environmental controls in Table 1 and Table 2. We also refer to the weather conditions and Fig. 2 in the Discussion. While we agree that the paper is not about the weather conditions at the site, we thus believe that the figure should not be removed to the Supplementary Material.

In Figure 6 it is unclear, which four locations are depicted within each panel, and this information is not in the Methods section of the main text. It is also unclear where the sediment/water interface is located in Figure 6 and how deep the water is in each

location. This information would be valuable for interpreting the discussion surrounding Figure at the end of page 12. Also I assume that these data [are] from the peepers and not the Vaisala probes, but it would be good to indicate this within the figure caption.

– We completely agree with these suggestions and suggest adding this information to the revised version of the paper.

Minor points Table S1: Reference error in the caption.

– We will correct this error.

---

## Author Comment (AC2) · 16 Apr 2016

Reviewer 2:

The manuscript was generally well written, but many references to supplementary information was irritating to my opinion. I recommend to consider better ways to refer to the supplement and diminish it's role.

I found the results section hard to read, because it was loaded with numbers i.e. listing means, medians and errors of the flux rates and results of statistical tests. Overall, there were very many statistical comparisons, and I think that not all of them are necessary to test the hypotheses.

[Figure]

– We will carefully look through the results section and improve the readability as suggested, also by reducing the number of statistical comparisons.

The comparison of mid summer and end summer was not motivated and it is not related to a hypotheses. Was the question related to the algal mats? The role of algal mats on the seasonality of C-gas fluxes, mentioned in the discussion, could be an interesting theme if better included in the story.

– The reviewer is correct with this comment. The comparison of mid-summer (up to August 7th) and end of summer (after August 7th) did not follow an "a priori" hypothesis or objective but was the result of two periods with clearly differing dynamics of the fluxes. The distinction was thus operationally defined. The presence of the algal mat, which dissolved on August 12th following a major storm, may have played role in this difference. The storm fell in the week between sampling dates 7th and August 15th. The observation that fluxes behaved differently after the mat had disappeared is indeed very interesting (Figure S4) and could play a role elsewhere too. Yet we did not expect such a development beforehand. Our measurements regarding the effects of the mat were thus rather incidental and we feel that we cannot substantiate the discussion much more than already done in the manuscript.

Specific comments and technical corrections

P. 3, L 18: You could try to explain this (. . .).

– We agree and will add an explanation based on the provided reference and others.

p. 3, introduction. Acknowledge the pond work conducted in the Hudson Bay Lowlands.

– We agree that this work was crucial for the research and will acknowledge it.

p. 3, l. 27. Hypothesis II. Rewrite to keep the consistency in the style.

– We agree with this suggestion and will correct the style accordingly.

p. 4., l. 27. Is the supplementary information necessary?

– We think that the detailed description of the instrumental parameters and setting of the measurements is helpful for other scientists and can't see why it should do harm in the manuscript. We would thus like to keep it in the Supplementary Information.

p. 5., l. 3-4, 6. Give the sensor make and model in the main text. Please, specify how you arranged the ice packs for the cooling. Did the packages block irradiation inside the chambers?

– We will provide more detail in the revised version of the manuscript and provide a photograph of the set-up that also shows how the ice-packs were arranged.

p. 11, l. 5-6. Total flux? I.e. ebullition and flux summed? Add this information also into fig. 9.

– We here considered the total fluxes. We will add this information to the caption of Figure 9 and also in the text. We will also clarify this point for the data presented in Table 1 and 2.

p. 11, l. 8- This is the most interesting piece. Maybe compress the previous results section to make this stronger and report gases as CH4 and CO2 (now only CO2 equivalents). Could you do a spatial extrapolation too? How the pond and floating mat contribute within the whole peatland 'complex'

– We agree with the reviewer that this is an important information with regard to the role of ponds for the greenhouse gas budget of peatlands. However the data set is rather small for scaling up to the entire peatland complex, especially as the fluxes at the land-water interface are so variable. Also we captured only a relatively short period of time. We thus believe that the main focus should remain on the mechanisms and controls on fluxes in the current manuscript. We are currently doing more work on the ponds of Wylde Lake and may be able to present a somewhat more solid estimate in another manuscript in the future.

p. 11 Discussion. Discuss how the features of the floating mat could affect 'physically' to the flux'. How you secured that the measurement/measurer did not cause disturbance increasing bubble release? Is it possible that gases are trapped under the floating mat? Is the water under the mat enriched with the gases? Etc..

– We used a long wooden board floating on canisters on the pond-end (floating board-walk) to do our measurements and to minimize pressure on the ground. The other end was secured at the drier end of the floating mat. Furthermore we discarded all fluxes that appeared to be influenced by placing the chamber. As pointed out in the method section, we discarded all measurements where methane concentration increased sharply within the first 60 seconds of chamber placement. The measurement was then repeated, which may perhaps have led to underestimation of fluxes. On the open water the float with chamber was secured in one place by a couple of telescopic poles that were rigidly connected to the floating boardwalk. This way we avoided a constant drifting of the chamber when the algae mat was present. It seems possible that gas was trapped under the floating mat but we did not attempt to quantify the concentration under the mat, which in hindsight would have been a useful measurement to do. We thus think that we do not have enough quantitative information for an extensive discussion of these points in the discussion itself; rather we will add some more information to the methods section and images of the setup to the Supplementary Information. this way the reader can get a better picture of the way the data were obtained when wished for.

p. 11, l. 23-24. Please, include some of the references in the main document. Supplement does not work well in this way.

– We will move the more important references and numbers to the discussion, as requested.

p. 13, l. 19. Reference to literature instead of Supplementary.

– We agree and will add appropriate references here. The point was to refer to the vegetation assessment, which is contained in the Supplementary Information.

[Figure]

I did not check the reference list.

– We will check again.
* * *

---

## Author Response (AR1)

**Author's response - Revision notes bg-2015-594**

**Reviewer 1:**

Confusion about CO2 gradient method:  Here we compared the results of the gradient method with the chamber measurements at two different times, a) in September when the results were similar, and b) in July when fluxes obtained with the gradient method were overestimated.

*We clarified the wording of this statement in the revised version of the manuscript.*

The authors might add another sentence or two in the methods section to justify why they used the gradient technique and why fluxes and why they differ from chamber measurements.

*We used the gradient method coupled to infrared sensor measurements to obtain daily amplitudes of $CO_2$ concentrations and fluxes, which is difficult to do with chamber measurements.  We agree with the reviewer and added this motivation to the methods section.*

I believe that Figure 9 might be eliminated and that the information within it can be added in tables 1 and 2.

*As the reviewer rightly states, Tables 1 and 2 detail relationships between environmental drivers and $CO_2$ and $CH_4$ flux. The relationships consistently consist of correlations, however, which differs from the comparison of means with the Kruskal Wallis test. To reduce the number of figures, as requested, we thus suggest adding the four numbers and standard deviations of the fluxes to the text, and moving the figure itself, which essentially visualizes the distribution of flux values, to the Supplementary information.*

Figure might be added to the  Supplementary Material since it is not thoroughly discussed within the results and discussion.

*In the results we have an own section on the weather data, which is in our opinion crucial for evaluation of the data, in particular also because of the presentation of environmental controls in Table 1 and Table 2. We also refer to the weather*

*conditions and Fig. 2 in the Discussion. While we agree that the paper is not about the weather conditions at the site, we thus believe that the figure should not be removed to the Supplementary Material.*

In Figure 6 it is unclear, which four locations are depicted within each panel, and this information is not in the Methods section of the main text. It is also unclear where the sediment/water interface is located in Figure 6 and how deep the water is in each location.   This information would be valuable for interpreting the discussion surrounding Figure  at the end of page 12. Also I assume that these  data [are] from the peepers and not the Vaisala probes, but it would be good to indicate this within the figure caption.

*We completely agree with these suggestions and added additional information to the revised version of the paper.*

Minor points

Table S1: Reference error in the caption.

*We will correct this error.*

**Reviewer 2:**

The manuscript was generally well written, but many references to supplementary information was irritating to my opinion. I recommend to consider better ways to refer to the supplement and diminish it's role.

I found the results section hard to read, because it was loaded with numbers i.e. listing means, medians and errors of the flux rates and results of statistical tests. Overall, there were very many statistical comparisons, and I think that not all of them are necessary to test the hypotheses.

*We will carefully look through the results section and improve the readability, also by reducing the number of statistical comparisons.*

The comparison of mid summer and end summer was not motivated and it is not related to a hypotheses. Was the question related to the algal mats? The role of algal mats on the seasonality of C-gas fluxes, mentioned in the discussion, could be an interesting theme if better included in the story.

*The reviewer is correct with this comment.  The comparison of mid-summer (up to August 7$^{th}$) and end of summer (after August 7$^{th}$) did not follow an a priori hypothesis*

*or objective but was the result of two periods with clearly differing dynamics of the fluxes.  The distinction is thus operationally defined. The presence of the algal mat, which dissolved on August 12th following a major storm, may have played role in this difference. The storm fell in the week between sampling dates 7th and August 15th. The observation that fluxes behaved differently the mat disappeared is indeed very interesting (Figure S4) and could play a role elsewhere too.  Yet we did not expect such a development beforehand. Our measurements regarding the effects of the mat were thus rather incidental and we feel that we cannot substantiate the discussion much more than already done in the manuscript.*

Specific comments and technical corrections

P. 3, L 18: You could try to explain this (…).

*We agree and will add an explanation based on the provided reference and others.*

p. 3, introduction. Acknowledge the pond work conducted in the Hudson Bay Lowlands.

*We agree that this work was crucial for the research. We added another reference and in addition to the papers that are cited were conducted in the Hudson Bay Lowlands.*

p. 3, l. 27. Hypothesis II. Rewrite to keep the consistency in the style.

*We agree with this suggestion and will corrected the style accordingly.*

p. 4., l. 27. Is the supplementary information necessary?

*We think that the detailed description of the instrumental parameters and setting of the measurements is helpful for other scientists and can't see why it should do harm in the manuscript.  We would thus like to keep it in the Supplementary Information.*

p. 5., l. 3-4, 6. Give the sensor make and model in the main text. Please, specify how you arranged the ice packs for the cooling. Did the packages block irradiation inside the chambers?

*We provided more detail in the revised version of the manuscript and provide a photograph of the set-up.*

p. 11, l. 5-6. Total flux? I.e. ebullition and flux summed? Add this information also into fig. 9.

*We here considered the total fluxes. We added this information to the caption of Figure 9 and also in the text. We also clarified this point for the data presented in Table 1 and 2.*

p. 11, l. 8- This is the most interesting piece. Maybe compress the previous results section to make this stronger and report gases as CH4 and CO2 (now only CO2 equivalents).
Could you do a spatial extrapolation too? How the pond and floating mat contribute within the whole peatland 'complex'

*We agree with the reviewer that this is important information with regard to the role of ponds for the greenhouse gas budget of peatlands. However the data set would be rather small for scaling up to the entire peatland complex, especially as the fluxes at the land-water interface are so variable. Also we captured only a relatively short period of time. We thus believe that the main focus should remain on the mechanisms and controls on fluxes in the current manuscript. We are currently doing more work on the ponds of Wylde Lake and may be able to present a somewhat more solid estimate in another manuscript in the future.*

p. 11 Discussion. Discuss how the features of the floating mat could affect 'physically' to the flux'. How you secured that the measurement/measurer did not cause disturbance increasing bubble release? Is it possible that gases are trapped under the floating mat? Is the water under the mat enriched with the gases? Etc..

*We used a long wooden board floating on canisters on the pond-end (floating boardwalk) to do our measurements and to minimize pressure on the ground. The other end was secured at the drier end of the floating mat. Furthermore we discarded all fluxes that appeared to be influenced by placing the chamber. As pointed out in the method section, we discarded all measurements where methane concentration increased sharply within the first 60 seconds of chamber placement. The measurement was then repeated, which may have perhaps led to underestimation of fluxes. On the open water the float with chamber was secured in one place by a couple of telescopic poles that were rigidly connected to the floating boardwalk. This way we avoided a constant drifting of the chamber when the algae mat was present.*
*It seems possible that gas was trapped under the floating mat but we did not attempt to quantify the concentration under the mat, which in hindsight would have been a useful measurement to do.*
*We thus think that we do not have enough quantitative information for an extensive discussion of these points in the discussion itself; rather we added some more information to the methods section and images of the setup to the Supplementary Information for the reader to get a better picture of the way the data were obtained.*

p. 11, l. 23-24. Please, include some of the references in the main document. Supplement
does not work well in this way.

*We cite the more important references in the discussion, as requested and only refer to the supplement in a summarizing way.*

p. 13, l. 19. Reference to literature instead of Supplementary.

*We agree and deleted the reference to the Supplementary Information.*

I did not check the reference list.

*We will check again.*

[revised manuscript text omitted]